# Amino acid dependent formaldehyde metabolism in mammals

Matthias Pietzke[1], Guillermo Burgos-Barragan[2,5], Niek Wit[2], Jacqueline Tait-Mulder[1], David Sumpton [1], Gillian M. Mackay[1], Ketan J. Patel[2,3] & Alexei Vazquez [1,4✉]

Aldehyde dehydrogenase class 3, encoded by *ADH5* in humans, catalyzes the glutathione dependent detoxification of formaldehyde. Here we show that ADH5 deficient cells turn over formaldehyde using alternative pathways starting from the reaction of formaldehyde with free amino acids. When mammalian cells are exposed to formaldehyde, the levels of the reaction products of formaldehyde with the amino acids cysteine and histidine - timonacic and spinacine - are increased. These reactions take place spontaneously and the formation of timonacic is reversible. The levels of timonacic are higher in the plasma of $Adh5^{-/-}$ mice relative to controls and they are further increased upon administration of methanol. We conclude that mammals possess pathways of cysteine and histidine dependent formaldehyde metabolism and that timonacic is a formaldehyde reservoir.

[1] Cancer Research UK Beatson Institute, Switchback Road, Bearsden, Glasgow G61 1BD, UK. [2] MRC Laboratory of Molecular Biology, Francis Crick Avenue, Cambridge CB2 0QH, UK. [3] University of Cambridge, Department of Medicine, Addenbrooke's Hospital, Cambridge CB2 2QQ, UK. [4] Institute of Cancer Sciences, University of Glasgow, Switchback Road, Bearsden, Glasgow G61 1QH, UK. [5] Present address: Meyer Cancer Center, Weill Cornell Medicine, New York, NY 10065, USA. ✉email: alexei.vazquezvazquez@glasgow.ac.uk

Formaldehyde is a highly reactive molecule and a known carcinogen[1]. The cancer risk associated with environmental exposure to formaldehyde has been extensively studied[2]. There is also an increased appreciation for the potentially harmful effects of formaldehyde generated by our endogenous metabolism. Endogenous formaldehyde is a source of DNA damage to haematopoietic cells[3] and formaldehyde accumulation has been associated with neurodegeneration[4].

Formaldehyde is formed in mammalian cells from demethylation reactions, the oxidative breakdown of folates and the metabolism of methanol, methylamine and adrenaline[5,6]. Intracellular formaldehyde is turned over to formate dependent on the activity of two different enzymes: mitochondrial aldehyde dehydrogenase and cytosolic aldehyde dehydrogenase class 3[7,8].

Mitochondrial aldehyde dehydrogenase is encoded by the *ALDH2* gene in humans. The gene product, ALDH2, localises to the mitochondria where it catalyses the detoxification of aldehydes. ALDH2 has a half-saturation constant for formaldehyde of 320 μM[9]. In isolated hepatocytes, about 25% of the formaldehyde turnover is dependent on the activity of ALDH2 when exposed to 200 μM formaldehyde, increasing to 32% when exposed to 1 mM formaldehyde[10].

Aldehyde dehydrogenase class 3 catalyses the glutathione-dependent detoxification of formaldehyde in the cytosol[11]. This enzyme is encoded by the gene *ADH5* in humans, *Adh5* in mice and it is ubiquitously expressed across tissues. ADH5-deficient cells are more sensitive to formaldehyde exposure than their parental controls[5]. The 50% lethal dose of formaldehyde goes down from 0.2 g/kg in *Adh5*-competent mice to 0.13 g/kg in mice with homozygous deletion of *Adh5*[12].

Experiments with formaldehyde in solutions containing amino acids have shown that formaldehyde can react with amino acids[13–15], indicating the existence of additional pathways of formaldehyde metabolism. Formaldehyde reacts quickly and forms stable products with the amino acids cysteine and histidine in a wide range of pH[15].

Here we address the relevance of these reactions in mammalian cells. We develop analytical assays, cell culture protocols and in vivo protocols to investigate formaldehyde metabolism. Using untargeted metabolomics, we uncover the accumulation of the formaldehyde adducts with the amino acids cysteine and histidine in cells. We validated the formation of these molecules in mice treated with methanol, a formaldehyde precursor. We discuss the relevance of these observations for our understanding of endogenous formaldehyde metabolism in mammals.

## Results

### Formaldehyde quantification with isotope resolution. The first challenge in investigating formaldehyde metabolism is to quantify formaldehyde and its stable isotope fractions. We developed a protocol to quantify formaldehyde in aqueous samples, using as a starting point a derivatization with O-(2,3,4,5,6-pentafluorobenzyl) hydroxylamine (PFBHA)[16]. PFBHA and the formaldehyde-PFBHA adduct are separated and detected by gas chromatography and mass spectrometry (GC–MS). PFBHA elutes at 4.2 min and formaldehyde-PFBHA at 3.5 min (Fig. 1a). The combined GC–MS chromatogram exhibits multiple peaks corresponding with different fragments of the PFBHA and formaldehyde-PFBHA molecules. The fragment with mass charge ratio (m/z) 181 has the highest intensity, and is suited for detection of total formaldehyde with the highest sensitivity (Fig. 1b). However, it does not contain the carbon atom from the formaldehyde molecule, and therefore it is not suitable for differentiating formaldehyde molecules with different stable isotopes. Although the fragments with m/z 195–197 exhibit a

lower intensity, their peak is proportional to the formaldehyde concentration in the range between 2 and 100 μM, which is sensitive to the presence of different stable isotopes in the formaldehyde carbon (Fig. 1c; Supplementary Fig. 1). We developed a deconvolution method that estimates the concentration of $^{12}$C- and $^{13}$C-formaldehyde using as input the peak areas of the m/z 195–197 fragments and $^2$H$_2$-formaldehyde as an internal standard ('Methods'). The method is capable of resolving the isotope composition of mixtures of $^{12}$C- and $^{13}$C-formaldehyde in a quantitative manner (Fig. 1d).

**Formaldehyde metabolism in cell cultures.** The second challenge to investigate formaldehyde metabolism is its reactivity. To illustrate this point, we quantified the concentration of formaldehyde in cell-culture medium (IMDM + 10% serum), with or without cells, starting from an initial concentration of 40 μM formaldehyde. In the absence of cells, formaldehyde is turned over to half its initial concentration in about 24 h (Fig. 1e). The formaldehyde turnover is much faster in the presence of cells. The cell cultures completely depleted 40 μM formaldehyde in 12 h (Fig. 1e). From these data, we conclude that direct formaldehyde supplementation is not a suitable experimental design to investigate formaldehyde metabolism in the steady state.

To search for alternatives, we fed HAP1 cells the formaldehyde precursors [methyl-$^{13}$C]-methionine, $^{13}$C-methanol and $^{13}$C-methylamine. HAP1 cells convert formaldehyde to formate, and formate is released to the extracellular medium[5]. We consequently used the presence of $^{13}$C-formate in media as readout for the substrate conversion to formaldehyde. $^{13}$C-methylamine was the only substrate leading to $^{13}$C-formate detection, with a production rate higher than that of $^{13}$C-serine (Fig. 1f), the natural source of formate in HAP1 cells[17]. The lack of incorporation of $^{13}$C-from methanol to formate also indicates the absence of significant alcohol dehydrogenase and catalase activity in HAP1 cells, the two major enzymes metabolising methanol, in mammalian cells. This does not exclude $^{13}$C-methanol as a suitable substrate to investigate formaldehyde metabolism in hepatocytes where these two enzymes are highly expressed.

Methylamine (CH$_3$-NH$_2$) is broken down by the semicarbazide-sensitive amine oxidase (SSAO) into formaldehyde, ammonia and H$_2$O$_2$[18–20]. SSAO is present in the serum, which represents 10% of our cell-culture medium. We hypothesised that the formation of formaldehyde is taking place in the cell culture medium in a SSAO-dependent manner. To test this hypothesis, we performed additional experiments with cell-culture medium without cells. There was no significant formaldehyde production in the medium without serum or in serum containing medium supplemented with 20 nM of the SSAO inhibitor PXS 4728 A (Fig. 1g). These data confirm that the conversion of methylamine to formaldehyde is dependent on the enzymatic activity of serum SSAO.

While conducting these experiments, we uncovered that the effective formaldehyde concentration in the cell-culture medium is affected by how many cells we seeded. This could be said about any metabolite taken up by cells, but in the case of formaldehyde, the effect is more pronounced. The steady-state concentration of formaldehyde is determined by the methylamine concentration, the number of cells in the culture and the genetic background. Assuming first-order kinetics of formaldehyde turnover, we developed the formaldehyde balance equation

$$f = (k_{media} + k_{cell}N)[\text{formaldehyde}], \qquad (1)$$

where $f$ is formaldehyde production rate by SSAO, $k_{media}$ and $k_{cells}$ are the first-order kinetic constants of formaldehyde turnover by

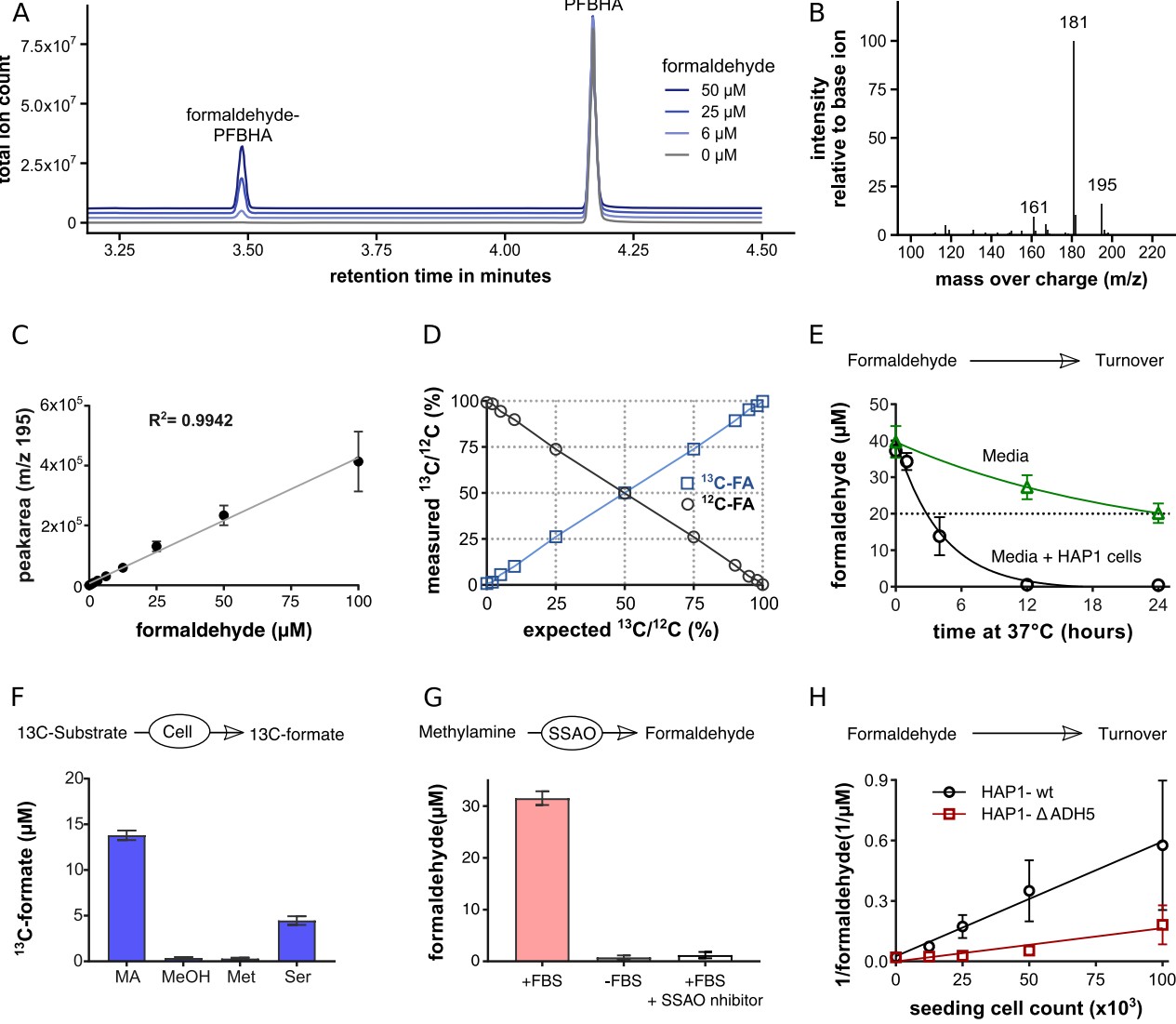

**Fig. 1 Formaldehyde quantification and kinetics. a** Gas chromatography of IMDM medium spiked with different amounts of formaldehyde, after PFBHA derivatisation. The PFBHA and formaldehyde–PFBHA peaks are indicated. **b** Mass spectra of formaldehyde–PFBHA. **c** Peak area of $m/z = 195$ for different formaldehyde concentrations. The line is a linear fit to the data. **d** Measured $^{13}C/^{12}C$ formaldehyde ratio for custom-made $^{13}C/^{12}C$ formaldehyde mixtures. The lines represent the measurement coinciding with the expectation. **e** Formaldehyde turnover by cell-culture medium (IMDM + 10% serum) without and with HAP1 cells, incubated at 37 °C. **f** $^{13}C$-formate concentration in HAP1 cell-culture medium with 200 μM of $^{13}C$-methylamine (MA), $^{13}C$-methanol (MeOH), [methyl-$^{13}C_1$]-methionine (Met) and [$^{13}C_3$]-serine (Ser). **g** $^{13}C$-formaldehyde quantification in IMDM medium incubated with 200 μM $^{13}C$-methylamine for 24 h at 37 °C, with or without serum (FBS), or with serum and 20 nM of the SSAO inhibitor PXS 4728A. **h** Free $^{13}C$ formaldehyde in cell-culture supernatant of HAP1-WT and -ΔADH5 cells seeded at different densities and incubated for 24 h with 400 μM $^{13}C$-methylamine. Statistics: points without error bars represent independent experiments. Bar heights and error bars represent the average and the standard deviation over three independent experiments.

the culture medium and the cells, respectively and $N$ is the number of cells. From Eq. (1) it follows that

$$\frac{1}{[\text{formaldehyde}]} = \frac{k_{\text{media}}}{f} + \frac{k_{\text{cell}}}{f} N. \quad (2)$$

That is, the inverse of the formaldehyde concentration scales linearly with the seeded cell density. We tested Eq. (2) in cell cultures of HAP1 cells (HAP1-WT) and HAP1 cells with genetic inactivation of *ADH5* (HAP1-ΔADH5). We seeded different counts in methylamine-containing medium and quantified the formaldehyde concentration in the cell-culture medium. For both cell lines, we observed a linear dependence as deduced from the steady-state equation (Fig. 1h). The intercept with the Y axis

($k_{\text{media}}/f$) is independent of what cell type is present in the culture. This is validated by the experimental data, where the lines fitted to the HAP1-WT and HAP1-ΔADH5 data intercept at about the same point with the Y axis. In contrast, the slope ($k_{\text{media}}/f$) is cell-type specific. As measured, the slope for HAP1-WT cells is larger than for HAP1-ΔADH5 cells (Fig. 1h), indicating that WT cells turn over formaldehyde at a faster rate per cell.

**Formation of formaldehyde adducts with amino acids in cells.** While HAP1-ΔADH5 turn over formaldehyde at a lower rate, they still follow Eq. (2), indicating that they have a cell-dependent mechanism of formaldehyde turnover that is independent of ADH5. To identify the metabolic pathways mediating this

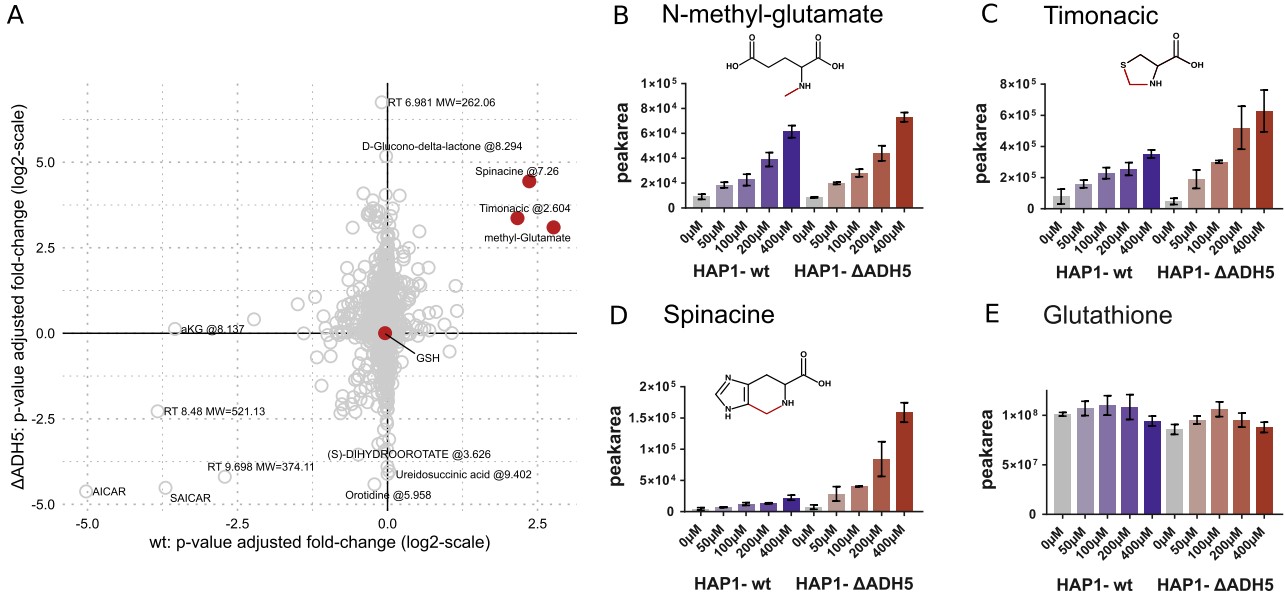

**Fig. 2 Identification of putative formaldehyde-reaction products in cells. a** Scatterplot reporting the P-value-scaled fold changes of metabolite levels in HAP1-WT and -ΔADH5 cell cultures treated for 24 h with 400 μM methylamine, as identified by the untargeted analysis. **b–e** Peak areas quantified by the targeted analysis of selected metabolites with increasing concentrations of methylamine, in HAP1-WT and -ΔADH5 cells. The red highlights the putative bonds between the formaldehyde carbon and the candidate substrate. Statistics: **b–e** Bar heights represent the average, and the error bars the standard deviation from three wells in one representative experiment.

cell-dependent and ADH5-independent formaldehyde turnover, we performed untargeted LC–MS analysis of intracellular metabolites in HAP1-WT and -ΔADH5 cells exposed to methylamine concentrations from 0 to 400 μM. We identified three compounds that are elevated in both wild-type and ADH5-deficient cells exposed to methylamine: methyl-glutamate, timonacic and spinacine (Fig. 2a). Methyl-glutamate is the reaction product of methylamine and glutamate. Timonacic, also known as thiazolidine-4-carboxylic acid or thioproline, is the reaction product of formaldehyde and cysteine[21]. Spinacine is the reaction product of formaldehyde with histidine[22]. Methyl-glutamate and timonacic increase in both cell lines in a dose-dependent manner with increasing the concentration of methylamine (Fig. 2b). The increase in spinacine is more pronounced in the ΔADH5 than in the WT cells (Fig. 2c, d). Surprisingly, we did not observe any significant difference in the levels of glutathione (GSH, Fig. 2e).

The identification of timonacic and spinacine prompted us to investigate their metabolism. Formaldehyde reacts spontaneously with cysteine and histidine[13–15], and timonacic is further converted into N-formylcysteine by bacteria and isolated mitochondria[23–26]. Putting these data together, we reconstructed the pathways depicted in Fig. 3a. To start testing the relevance of these pathways, we performed experiments with phosphate-buffered saline solution containing 1.25 mM $^{13}$C-formaldehyde and 2.50 mM of cysteine, histidine or glutathione. When cysteine was present, we detected a rapid and complete depletion of formaldehyde within 2 h, together with the formation of $^{13}$C-timonacic and $^{13}$C-N-formylcysteine (Fig. 3b–d, circles). When histidine was present, there is a slight and linear decrease in the formaldehyde concentration, together with a linear increase in $^{13}$C- spinacine (Fig. 3b, e, triangles up). The linearity of these plots is evidence that the histidine + formaldehyde reaction has not reached saturation within 6 h. Finally, when glutathione was present, we did not observe a significant depletion of formaldehyde or glutathione (Fig. 3f, squares), the formation of $^{13}$C-

hydroxymethyl-glutathione or the reported adduct of formaldehyde and glutathione[27,28].

We also analysed phosphate-buffered saline solutions containing 1.25 mM $^{13}$C-formaldehyde, and equal amounts (0.83 mM) of cysteine, histidine and glutathione. In this case, formaldehyde was depleted from 125 μM to about 50 μM within the first 2 h (Fig. 3b, triangles down). This fast kinetics matches what with what was observed for cysteine alone, and it is corroborated by the saturation in the formation of $^{13}$C-timonacic and $^{13}$C-N-formylcysteine (Fig. 3c, d, triangles down). Here again we observed the linear increase in the formation of $^{13}$C-spinacine and a lack of glutathione depletion (Fig. 3e, f, triangles down). Based on these data, in solution, cysteine has a much higher reactivity towards formaldehyde than histidine and glutathione, and histidine has more reactivity than glutathione.

If all the 0.83 mM of cysteine present in the mixture reacted with formaldehyde, then, we would have expected a drop from 1.25 to 1.25 – 0.83 = 0.42 mM $^{13}$C-formaldehyde, Instead, we observed a drop to 0.5 mM $^{13}$C-formaldehyde (Fig. 3b, triangles down). This mismatch indicates that formaldehyde is not fully depleted in mixtures containing cysteine. One possibility is that the formation of timonacic is reversible. To test the reversibility, we incubated $^{12}$C-timonacic with $^{13}$C-formaldehyde in tenfold excess in PBS solution (pH 7.4). After 12 h, the sample switched from the initial $^{12}$C-timonacic to almost 100% $^{13}$C-timonacic (Fig. 3g). Since cysteine was not provided in pure form, the only explanation is that $^{12}$C-timonacic was converted back to $^{12}$C-formaldehyde and cysteine, and free cysteine quickly reacted with the excess $^{13}$C-formaldehyde to form $^{13}$C-timonacic. In our experiments, in PBS solution at pH 7.4, it takes 24 h to fully convert $^{12}$C-timonacic to $^{13}$C-timonacic. In contrast, in a water solution at pH 7.0, the conversion of $^{12}$C- timonacic to $^{13}$C-timonacic is not complete at 48 h[15]. Nevertheless, in both conditions, there is reversible release of formaldehyde from timonacic.

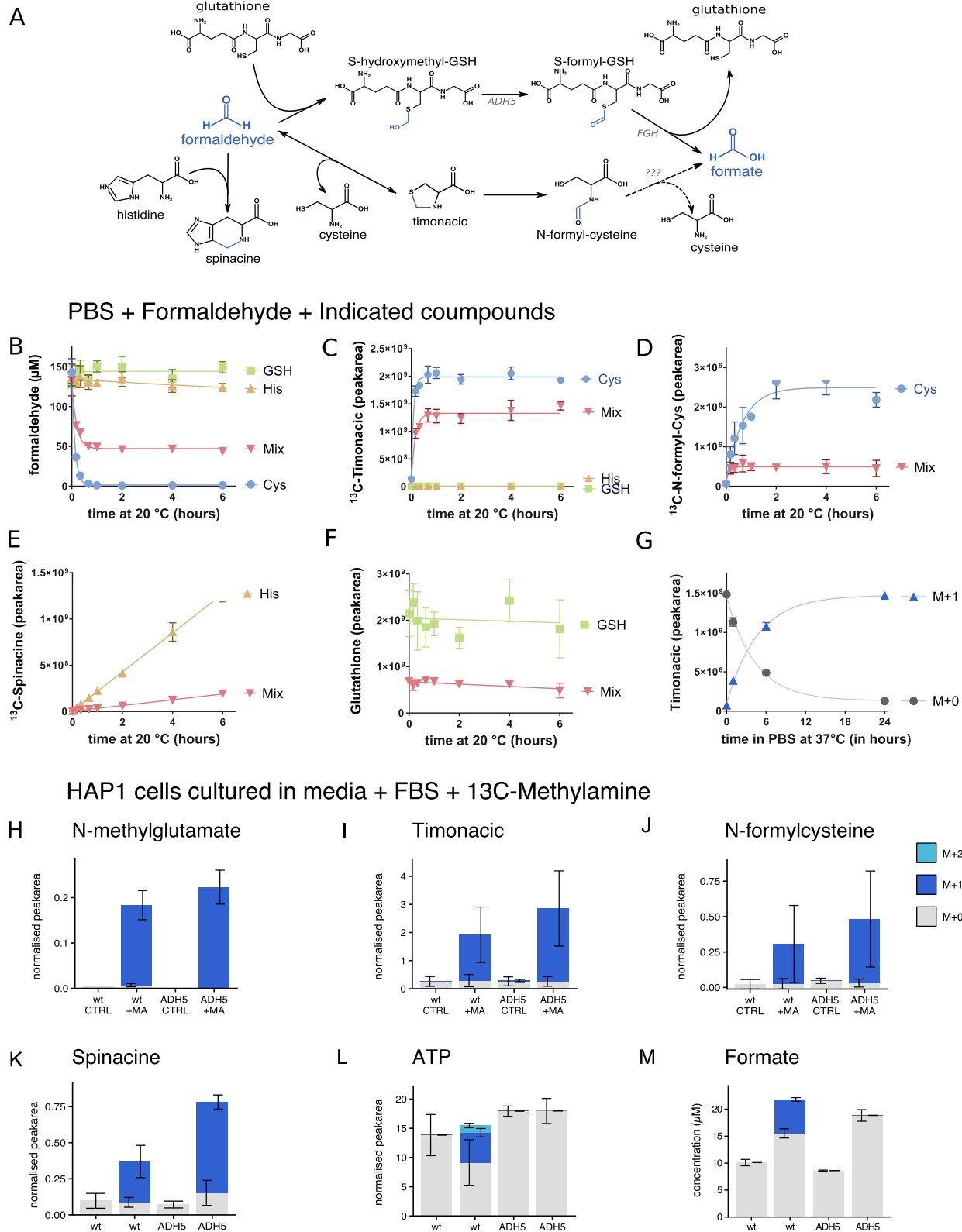

**Fig. 3 Chemistry of formaldehyde in solution and cells. a** Pathways of formaldehyde metabolism. **b**–**f** Metabolite levels in solutions of 1.25 mM [13]C-formaldehyde and 2.50 mM of the compound listed in the legend. The mix contains 0.83 mM of cysteine, histidine and gluthathione. **g** Conversion of [12]C- to [13]C-timonacic starting from a solution of 2.5 mM [12]C- timonacic and 25 mM [13]C-formaldehyde. **h**–**m** Levels of intracellular metabolites (normalised to inositol) and extracellular formate in HAP1-WT (wt) and HAP1-ΔADH5 (ADH5) cell cultures containing 400 μM [13]C-methylamine. M + 0, 1, 2 stand for molecules containing 0, 1, 2 [13]C-atoms, respectively. Statistics: bar heights and error bars represent the mean and the standard deviation from three independent experiments. Curves are fitted using one-phase decay, except for 1e and 1f where a linear fit was used (GraphPad Prism 7.02).

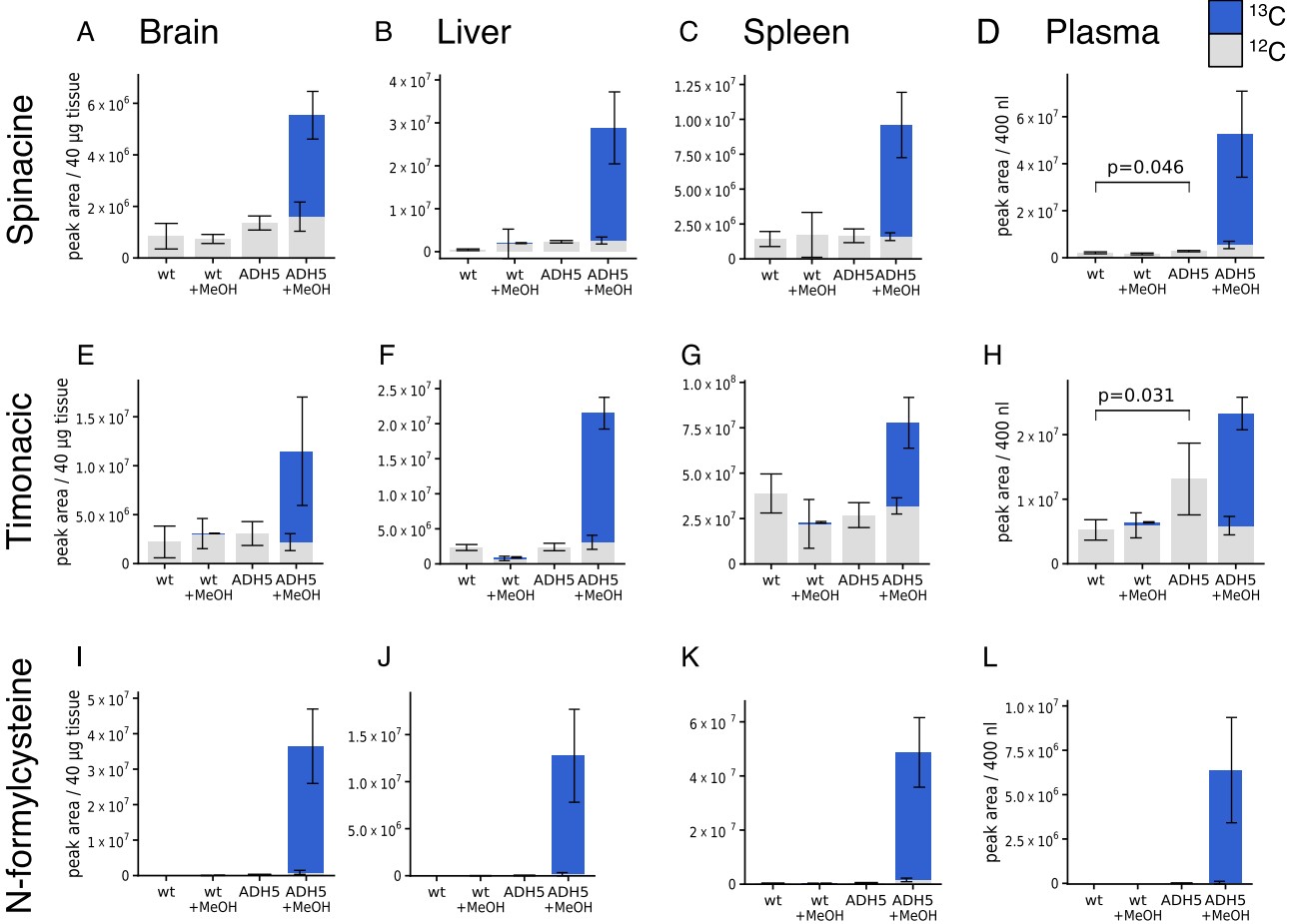

**Fig. 4 Formation of formaldehyde adducts in mouse tissues. a–l** Levels of spinacine, timonacic and N-formylcysteine in tissues (peak area/mg) and blood plasma (peak area/40 mL) of $Adh5^{+/+}$ and $Adh5^{-/-}$ C57BL/6J mice, measured by LC–MS. Mice received an intraperitoneal injection of vehicle (CTRL) or $^{13}C$-methanol (MeOH, 3 g/kg of body weight). Grey bars indicate the naturally occurring $^{12}C$ metabolite, and blue bars the $^{13}C$ metabolite formed from $^{13}C$-methanol. *Statistics:* bar heights represent the mean, error bars the standard deviation of four mice (CTRL) and six mice (MeOH)and statistical significances using a one-tail unequal variance *t* test.

To corroborate the occurrence of these reactions in cells, we cultured HAP1-WT and ΔADH5 cells in medium containing 400 μM of $^{13}C$-methylamine and traced the incorporation of $^{13}C$ into intracellular metabolites and extracellular formate. We observed the formation of $^{13}C$-methyl-glutamate, $^{13}C$-timonacic, $^{13}C$-N-formylcysteine and $^{13}C$ spinacine in both WT and ΔADH5 cells (Fig. 3h–k). We did not detect $^{13}C$-hydroxymethyl-glutathione or $^{13}C$-S-formyl-glutathione. We observed the incorporation of $^{13}C$ into intracellular ATP and extracellular formate (Fig. 3l, m), but only in the WT cells, corroborating that the conversion of formaldehyde to formate is ADH5 dependent in HAP1 cells[5].

**Formation of formaldehyde adducts with amino acids in mice.** To investigate the occurrence of these reactions in whole organisms, we conducted metabolomic analyses of tissues and blood plasma of C57BL/6J mice ($Adh5^{+/+}$) and C57BL/6J mice with a whole-body homozygous deletion of $Adh5$ ($Adh5^{-/-}$). We detected naturally occurring $^{12}C$-timonacic and $^{12}C$- spinacine in all tissues and plasma (Fig. 4a–h). The $^{12}C$-timonacic levels are significantly elevated in the plasma of $Adh5^{-/-}$ mice relative to the plasma of $Adh5^{+/+}$ mice (Fig. 4h). This observation is consistent with a previous study reporting the elevation of $S$-nitrosothiols in the red blood cells of C57BL/6J $Adh5^{-/-}$ mice relative to $Adh5^{+/+}$ control mice. ADH5 also has $S$-nitrosoglutathione

reductase (GSNOR) activity, thus explaining the accumulation of $S$-nitrosothiols[29]. We also noted an elevation of $^{12}C$-spinacine in the plasma of $Adh5^{-/-}$ mice relative to the plasma of $Adh5^{+/+}$ mice, although the signal is not as significant as for $^{12}C$-timonacic (Fig. 4d).

To challenge formaldehyde metabolism beyond basal levels, we also investigated mice injected intraperitoneally with $^{13}C$-methanol. $^{13}C$-methanol is converted in the liver and other tissues into $^{13}C$-formaldehyde, which then can react with amino acids. We detected a massive increase of $^{13}C$ timonacic, $^{13}C$-N-formylcysteine and $^{13}C$-spinacine in all tissues and plasma of $Adh5^{-/-}$ mice, but not of $Adh5^{+/+}$ mice (Fig. 4). N-formylcysteine was not detected in any combination of genetic background and condition other than $Adh5^{-/-}$ mice challenged with methanol (Fig. 4i–l). The switch-like behaviour in the detected levels of N-formylcysteine indicates that N-formylcysteine turnover has a maximum capacity. When the production of N-formylcysteine exceeds that capacity, N-formylcysteine accumulates. One hypothesis is that the turnover of N-formylcysteine is enzymatic, and that the enzymatic capacity of this hypothetical enzyme is exceeded in $Adh5^{-/-}$ mice challenged with methanol.

We noted some differences between the in vitro and in vivo experiments. We observed the formation of $^{13}C$ adducts of

formaldehyde with cysteine and histidine in wild-type HAP1 cells exposed to [13]C-methylamine (Fig. 3i, k), but not in wild-type mice receiving [13]C-methanol (Fig. 4e–l). One possible explanation is that we achieve a higher dose of intracellular formaldehyde in the in vitro experiments. Indeed, we observed the formation of [13]C adducts of formaldehyde with cysteine and histidine in $Adh5^{-/-}$ mice challenged with [13]C-methanol.

## Discussion

These observations highlight the role of cysteine and histidine in the metabolism of formaldehyde. These amino acids react spontaneously with formaldehyde in aqueous solution and at a rate faster than the reaction of formaldehyde with glutathione. The spontaneous reaction product of cysteine and formaldehyde—Timonacic—is more soluble than cysteine and less reactive than formaldehyde, which facilitates its transports across cells. The reaction of cysteine with formaldehyde is reversible. Timonacic is therefore a reservoir of formaldehyde that could be formed at some tissues, enters the circulation and dissociates back to cysteine and formaldehyde. In that sense, timonacic can contribute to the whole-body distribution of cysteine and formaldehyde. Since formaldehyde is converted to formate in a ADH5-dependent manner, timonacic may also contribute to the whole-body balance of formate, the canonical one-carbon unit[6]. Given that timonacic is detectable with LC–MS, this provides a precious analytical tool to investigate the elusive biochemistry of endogenous formaldehyde in mammals.

Further work is required to establish the route of formation and turnover of N-formylcysteine in mammalian cells. We do not have enough data to discriminate between the formation of N-formylcysteine from timonacic or from the direct reaction of formaldehyde with the amine group of cysteine. Furthermore, the dramatic increase of [13]C-N-formylcysteine, but not [13]C-timonacic, in $Adh5^{-/-}$ mice treated with [13]C-methanol, points to a bottleneck in the turnover of N-formylcysteine.

We note that our observations do not challenge the role of the glutathione and ADH5-dependent metabolism of formaldehyde to formate. Cell lines expressing ADH5 turn over formaldehyde to formate, as expected from the glutathione and ADH5-dependent metabolism of formaldehyde[5,6]. We have not been able to obtain reliable quantifications of the expected intermediate metabolites hydroxymethyl-glutathione and formyl-glutathione, but that may just be a consequence of their rapid turnover or chemical instability. Instead, the formaldehyde reactions with cysteine and histidine are alternative routes of formaldehyde metabolism.

It remains to be determined what is the endogenous concentration of formaldehyde in blood and tissues. Our attempt to quantify formaldehyde in plasma and tissues from mice returned low μM values, in the range of the detection limit of our formaldehyde quantification protocol. Measurements using high-performance liquid chromatography and fluorescence detection yield formaldehyde concentrations in cells and normal tissues around 10 μM[30], in agreement with our expectation of low μM. However, the same technique yields formaldehyde concentrations above 40 μM in the hippocampus[31]. Based on these observations, we expect a normal physiological concentration of formaldehyde in the low μM range.

Previous studies have reported associations between increased formaldehyde levels in the brain and neurodegeneration[4,32]. Toxicology studies in mice indicate that timonacic causes central nervous system toxicity at a dose of 125 mg/kg or more[33]. Whether administered timonacic caused the toxicity directly or acted as a formaldehyde-delivery system is unclear. Mammals exposed to formaldehyde doses in the order of 10 mg/kg/day manifest symptoms of Alzheimer's disease at the phenotypic and molecular level[34,35]. These toxicology data provide a causal link between timonacic/formaldehyde and central nervous system toxicity. It remains to be determined whether timonacic plays a role in the distribution of formaldehyde from distant sites to the brain, or it is simply formaldehyde generation in the brain.

## Methods

**Chemicals**. SSAO inhibitor PXS 4728A was obtained from Axon Medchem, [13]C-methanol from Eurisotope and [13]C-formaldehyde was originally purchased by the Patel lab and is a leftover from experiments previously performed together. Cell-culture medium is Iscove's Modified Dulbecco's Medium (IMDM), and other cell-culture components were obtained from Thermo Fisher Scientific. All other chemicals were obtained from Sigma-Aldrich.

**Cell culture and design of in vitro experiments**. HAP1-wt and HAP1-ΔADH5 cells were obtained from Patel's lab. Cells were cultivated in IMDM medium with 10% FBS, at 37 °C in a humidified atmosphere with 5% $CO_2$. All cells were tested regularly for mycoplasma contaminations.

For the experiments, cells were seeded with 100,000 cells per well into a 12-well plate (Gibco). The next day after seeding, the medium in each well was replaced with 1 ml of medium containing the indicated tracers. Cells were harvested 24 h after replacing the medium. For extracellular metabolites, the medium was collected, any leftover cells gently spun down (10 min at 500 g and 4 °C) and the medium was transferred to a new tube, kept at −80 °C before being aliquoted into different samples.

**LC–MS measurements**. Samples for LC–MS were prepared and measured as described in ref. [36]. Briefly, adherent cell cultures were cultivated in 12-well plates, washed once with ice-cold PBS, extracted with extraction solvent (ACN:MeOH: $H_2O$, 3:5:2) and separated on a pHILIC column and detected using an Orbitrap mass spectrometer. Parameters for identification of compounds can be found in Supplementary Table 2.

**Derivatisation of formate**. Formate was measured as benzyl derivative as described in ref. [37]. In total, 40 μl of the sample was mixed with 20 μl of internal standard (50 μl $d_2$ -formate), 10 μl of 1 M NaOH, 5 μl of benzyl alcohol and 50 μl of pyridine. Derivatisation was started by adding 20 μl of methylchloroformate while vortexing. Phase separation was established after adding 100 μl of MTBE and 200 μl of water, followed by 10 s of vortexing and 10 min of centrifugation (12.700 g at 4 °C). The upper (apolar) phase, containing the benzyl formate, was transferred to GC–MS vials with inserts, and measured on the same day.

**Derivatisation of formaldehyde**. PFBHA was dissolved in milli-Q water at 1 mg/ml and diluted to a working concentration of 0.1 mg/ml. $d_2$-formaldehyde was diluted in milli-Q water to a 50 μM solution. In total, 100 μl of sample was aliquoted into an Eppendorf tube, followed by the addition of 20 μl of a 50 μM $d_2$ -formaldehyde in milli-Q water as an internal standard. In all, 50 μl of PFBHA solution was added, and samples were shaken for 1 h at 20 °C. Samples were cooled down on ice for 5 min, followed by the addition of 100 μl of MTBE. Samples were then centrifuged for 10 min at maximum speed and 4 °C to establish a phase separation. Finally, ~50 μl of the upper apolar phase was transferred to a GC–MS vial with an insert, and measured on the same day.

**GC–MS detection of formate and formaldehyde**. Formate (as benzyl derivative) and formaldehyde (as FA-PFBHA derivative) were measured by GC–MS using the same set-up, but with different temperature gradients and MS parameters. An Agilent 7890B GC was used for the measurements, equipped with a Phenomenex ZB-1701 column (30 m × 0.25 mm × 0.25 μm) coupled to an Agilent 7000 triple-quad MS. The temperature of the inlet was 280 °C, the interface temperature was 230 °C and the quadrupole temperature was 200 °C, and the EI voltage was set to 60 eV. In all, 2 μl of the sample was injected and transferred to the column in split mode (1:25) with a constant gas flow through the column of 1 ml/min.

For formate, the oven temperature started at 60 °C, held for 0.5 min, followed by a ramp of 38 °C/min to 230 °C, which was held for another minute. The total run time was 6 min; the retention time of benzyl formate was 3.8 min. The mass spectrometer was operated in selected ion-monitoring (SIM) mode between 3.0 and 4.3 min with SIM masses of 136, 137 and 138 for M0, M + 1 and M + 2 (internal standard) formate, respectively.

For formaldehyde, the oven temperature started at 60 °C, followed by a ramp of 25 °C/min to 150 °C, and another ramp of 60 °C/min to 230 °C, which was held for 2 min. The total run time was 7 min; the retention time of FA-PFBHA was 3.5 min

and that of PFBHA was 4.175 min. The mass spectrometer was operated in Scan mode, mode between 2.6 and 6.8 min and with a scan range between 100 and 250 m/z.

**GC–MS data analysis**. Recorded data were processed using the MassHunter Software (Agilent). Integrated peak areas were extracted and used for further quantifications using in-house scripts, including the subtraction of natural isotope abundances, as described in ref. [37] for formate or below for formaldehyde.

**Isotopic deconvolution of formaldehyde**. The mass spectra of mixtures of $^{12}$C-, $^{13}$C- and $d_2$-formaldehyde revealed three fragments with m/z = 195, 197 and 197. The area of the three masses had two components: one corresponding to a fragment that has lost the formaldehyde group and another that retained the formaldehyde group. These different contributions to the measured peak areas were modelled as

$$
\begin{aligned}
A_0 &= (F_0 + F_1 + F_2)b_0 + (\alpha F_0)a \\
A_1 &= (F_0 + F_1 + F_2)b_1 + (\beta F_0 + \alpha F_1)a \quad , \\
A_2 &= (F_0 + F_1 + F_2)b_2 + (\chi F_0 + \beta F_1 + \alpha F_2)a
\end{aligned}
\tag{3}
$$

where $A_i$ are the measured peak areas, and $F_i$ are the concentrations of $^{12}$C-, $^{13}$C- and $^2$H$_2$-formaldehyde in the mixture, $b_i$ and $a$ are scaling factors translating amounts of substance to peak area and

$$
\alpha = (1-c)^n
\tag{4}
$$

$$
\beta = nc(1-c)^{n-1}
\tag{5}
$$

$$
\chi = \frac{n(n-1)}{2}c^2(1-c)^{n-2}
\tag{6}
$$

are correction factors for the mass shift due to the natural abundance of $^{13}$C in the derivatising molecule. In these equations, $n$ is the number of carbons in the derivatising molecule and $c$ is the natural abundance of $^{13}$C. Defining

$$
x_i = a\alpha F_i \quad i = 0, 1, 2
\tag{7}
$$

$$
\begin{aligned}
y_0 &= A_0 \\
y_1 &= A_1 - \frac{\beta}{\alpha}A_0 \\
y_2 &= A_1 - \frac{\beta^2}{\alpha^2}A_0 - \frac{\beta}{\alpha}A_1
\end{aligned}
\tag{8}
$$

we can transform Eq. (3) into the system of linear equations

$$
y = Mx
\tag{9}
$$

where

$$
M = \begin{bmatrix} c_0 + 1 & c_0 & c_0 \\ c_1 & c_1 + 1 & c_1 \\ c_2 & c_2 & c_2 + 1 \end{bmatrix}
\tag{10}
$$

$$
c_i = \frac{d_i}{d_3} \quad i = 0, 1, 2
\tag{11}
$$

$$
d_0 = b_0
\tag{12}
$$

$$
d_1 = b_1 - b_0\frac{\beta}{\alpha}
\tag{13}
$$

$$
d_2 = b_2 - b_1\frac{\beta}{\alpha} - b_0\frac{\beta^2}{\alpha^2}
\tag{14}
$$

$$
d_3 = a\alpha.
\tag{15}
$$

To estimate the model parameters $c_i$, we quantified different mixtures of $^{12}$C-, $^{13}$C- and $^2$H$_2$-formaldehyde and solved Eq. (9) for $d_i$. Specifically, using Eqs. (7)–(15), we obtained the following system of linear equations:

$$
y_k = N_k d,
\tag{16}
$$

where the index $k$ runs over the different formaldehyde mixtures used to train the model and

$$
N_k = \begin{bmatrix} F_k & 0 & 0 & F_{k,0} \\ 0 & F_k & 0 & F_{k,1} \\ 0 & 0 & F_k & F_{k,2} \end{bmatrix},
\tag{17}
$$

where $\left(F_{k,0}, F_{k,1}, F_{k,2}\right)$ are the known $^{12}$C-, $^{13}$C- and $^2$H$_2$-formaldehyde concentrations in the $k$ mixture and $F_k = F_{k,0} + F_{k,1} + F_{k,2}$. Using the mixtures reported in Supplementary Table 1, we solved Eq. (16) using the least-squares

method with non-negative variables obtained by the parameter estimates

$$
\begin{aligned}
c_0 &= 0.722978 \\
c_1 &= 0.153166 \\
c_2 &= 0.073341
\end{aligned}
\tag{18}
$$

To estimate the $F_0$ and $F_1$ concentrations in samples with a spiked concentration $F_2$ of $^2$H$_2$-formaldehyde standard, we solved Eqs. (7)–(8) by the least-squares method with non-negative variables, using as input the estimated c parameters in Eq. (18) and the measured peak areas. Putting all together, we arrive at the working system of equations

$$
y = Mx
\tag{19}
$$

with

$$
M = \begin{bmatrix} 1.722978 & 0.722978 & 0.722978 \\ 0.153166 & 1.153166 & 0.153166 \\ 0.073341 & 0.073341 & 1.073341 \end{bmatrix}
\tag{20}
$$

The formaldehyde isotopologue concentrations were then obtained as

$$
F_i = \frac{x_i}{x_2}F_2 \quad i = 0, 1.
\tag{21}
$$

**Untargeted identification of LC–MS data**. The data were analysed using Compound Discoverer software (Thermo Scientific v3.0). Retention times were aligned across all sample data files (maximum shift 2 min, mass tolerance 5 ppm). Unknown compound detection (minimum peak intensity 10$^6$) and grouping of compound adducts was carried out across all samples (mass tolerance 5 ppm, RT tolerance 0.2 min). Missing values were filled using the software's Fill Gap feature (mass tolerance 5 ppm, S/N tolerance 1.5). Feature identification was achieved by matching the mass and retention time of observed peaks to an in-house database generated using metabolite standards (mass tolerance 5 ppm, RT tolerance 2 min). In addition, ChemSpider node was used to suggest further possible peak annotations (search mass or formula, mass tolerance of 5 ppm, databases: HMDB, KEGG and BioCyc).

**Targeted identification of LC–MS data**. Compounds were identified using Tracefinder 4.1 (Thermo Scientific), comparing the exact mass and the retention time against an in-house compound database created with authentic standards. For timonacic, N-formylcysteine and glutathione, we purchased standards and verified their retention times. For spinacine, we have produced $^{13}$C-spinacine from $^{13}$C-formaldehyde and histidine and verified the spinacine retention time.

**Formaldehyde calibration (Fig. 1c)**. Formaldehyde was prepared in IMDM medium with a concentration of 400 μM and then further diluted twofold in a serial dilution down to 0.78125 μM (nine different concentrations). Then 100 μM of the sample was used for the derivatisation, as described above, without the addition of an internal standard. A set of three preparations was prepared independently and measured in parallel.

**Verification of isotopic deconvolution (Fig. 1d)**. $^{12}$C- and $^{13}$C-formaldehyde stocks were prepared with a concentration of 50 μM, and concentrations validated by similar peak area. $^{12}$C- and $^{13}$C- formaldehyde were mixed in known ratios (0, 2, 5, 10, 25, 50, 75, 90, 95, 98 and 100% $^{13}$C); 100 μl were collected, derivatised and measured as described above, including the addition of D$_2$ formaldehyde as internal standard (20 μl of a 50 μM solution). Preparation was repeated in triplicates.

**Formaldehyde stability (Fig. 1e)**. In total, 40 μM of $^{13}$C-formaldehyde was added to complete IMDM, including 10% FBS, and 1 ml of this medium was added to blank wells of a 12-well plate or to wells in which 100.000 HAP1-wt cells were seeded the day before the experiment. Plates were cultured in an incubator at 37 °C in a humidified 5% CO$_2$ atmosphere. At the indicated time points, the medium was harvested, cells spun down and the supernatant stored at −80 °C, before 100 μM of this were used for the formaldehyde quantification. The experiment was performed three times, with three replicates per experiment.

**Identification of potential formate precursors (Fig. 1f)**. HAP1-wt cells were seeded into 12-well plates, and at the following day, the medium was replaced with IMDM supplemented with no tracer or 200 μM of [$^{13}$C$_3$]-serine, [methyl-$^{13}$C$_1$]-methionine, $^{13}$C-methanol or $^{13}$C-methylamine. As the medium already contains 200 μM methionine and 400 μM serine, the relative proportion of the labelled species is 50 and 33%, respectively. The presence of methanol and methylamine can be neglected. Twenty-four hours after the medium change, the medium was harvested, and the concentration of formate was measured. The experiment was performed three times, with three replicates per experiment.

**Validation of SSAO-dependent FA formation in media (Fig. 1g).** IMDM with or without 10% FBS was supplemented with 200 µM of $^{13}$C-methylamine. One part of the FBS-containing medium was additionally supplemented with the SSAO inhibitor PXS 4728 A (final concentration 20 nM). The medium (1 ml) was incubated in triplicate in a 12-well plate at 37 °C in a humidified 5% $CO_2$ atmosphere for 24 h. Then the medium was collected, and the free formaldehyde in the medium was quantified. The experiment was performed three times independently.

**Formaldehyde consumption relative to cell count (Fig. 1h).** HAP1-wt or HAP1–ΔADH5 cells were seeded with 100000, 50000, 25000 or 12500 cells per well into 12-well plates. The next day, the medium was replaced with 1 ml of medium containing 400 µM $^{13}$C-methylamine; the same medium was added to wells without cells. After incubating for 24 h at 37 °C, the medium was collected, and formaldehyde was quantified. The experiment was performed three times independently, with three replicates per experiment.

**Untargeted metabolomics on methylamine treatment (Fig. 2).** HAP1-wt or HAP1–ΔADH5 cells were seeded with 100.000 cells per well into a 12-well plate. On the next day, the IMDM medium was replaced, and the IMDM medium was supplemented with methylamine in different concentrations from 0 to 400 µM, and incubated for a further 24 h. Intracellular metabolites were harvested and measured by LC–MS, followed by an untargeted data analysis using Compound Discoverer (Thermo Scientific). Further, selected compounds were identified and validated using targeted data processing. The experiment was performed at least three times, with three replicates in each experiment. The untargeted processing was performed on two of the repetitions, the targeted processing on three of the repetitions; one of the datasets is shown. Data were further processed and normalised with cell count using Metabolite AutoPlotter (https://mpietzke. shinyapps.io/AutoPlotter/).

To simultaneously compare all four conditions (combination of genetic background and treatment), we developed a '4 condition volcano plot'. First, we calculated log2 of the fold changes and –log10 of the P values for the replicate samples from the same experiment, between untreated control and the 400 µM methylamine condition, as one would do for a volcano plot. Then a correction factor was calculated, based on the –log10 of the P value, penalising noisy data with a high P value. A value of 2 was set as threshold, so everything with a P value smaller than or equal to 0.01 was treated identically. Then the –log10 of the P value was divided by this threshold (2), to calculate the penalty factor, resulting in a 1 for a P value of 0.01 (and smaller), 0.65 for a P value of 0.05, 0.5 for a P value of 0.1 and so on. Finally log2 of the fold change was multiplied with this penalty factor, and the adjusted fold changes were plotted, so compounds with a high P value were moved towards the centre of the plot, while compounds with a significant P value retained their original position.

**Reactivity of different compounds with formaldehyde (Fig. 3b–f).** Solutions of cysteine, glutathione or histidine or a mixture of these compounds were prepared in PBS, and a solution of $^{13}$C-formaldehyde was added, so that the final concentration of these compounds (or their mixture) was 2.5 mM, and with this twice the concentration of formaldehyde that was used with 1.25 mM, as this better reflects the situation in vivo, in which the concentration of formaldehyde should be lower than that of the amino acid traps. The mixture was shaken at 20 °C for up to 6 h. At the indicated time points after adding formaldehyde (0, 10, 20, 40, 60, 120, 240 and 360 min), a sample was taken and diluted 1:50 in extraction solvent for LC–MS measurements (concentration of compounds ~50 µM) or 1:10 in water for GC–MS-based detection of formaldehyde (highest concentration = 125 µM). Three individual preparations were prepared per compound and measured in parallel.

**Reversibility of timonacic formation (Fig. 3g).** A solution of $^{12}$C-timonacic in PBS (final concentration 2.5 mM) was mixed with tenfold excess of $^{13}$C-formaldehyde (final concentration 25 mM) and shaken at 37 °C. After 0, 1, 6 and 24 h, a sample was taken and diluted 1:50 in extraction solvent for LC–MS measurements (concentration of timonacic ~50 µM). Three individual preparations were prepared per compound and measured in parallel.

**$^{13}$C-methylamine tracing in vitro (Fig. 3h).** HAP1-wt or HAP1–ΔADH5 cells were seeded into 12-well plates at 100.000 cells per well into 12-well plates. The following day, the medium was replaced with medium containing no tracer or 400 µM $^{13}$C-methylamine. After 24 h of incubation, intracellular metabolites and medium formate were extracted and measured as described above.

**$^{13}$C-methanol tracing in vivo (Fig. 4).** Adh5$^{-/-}$ and controls were from a C57BL/6 background, and were generated as described previously (Pontel 2015 Mol. Cell). In individual experiments, all mice were matched for age and gender. All animal experiments undertaken in this study were conducted with the approval of the UK Home Office and the MRC Centre Ethical Review Committee.

The $^{13}$C-methanol tracing was performed as previously described[17].

Several tissues were harvested, shock-frozen and stored at −80 °C until further processing. Frozen tissues were cut into smaller pieces and weighed. In all, 1 ml of extraction solvent (−20 °C) was added per 20 mg of tissue sample. Tissues were

homogenised in a cooled Tissuelyser (Bertin Technology), with 7200 rpm, 20 s of shaking separated with a 20-s break. Samples were centrifuged to remove cellular debris, and the supernatant was transferred to LC–MS vials and stored at −80 °C until measurement.

## Data availability

All data relevant to this work are available in the paper and Supplementary Material.

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

## Acknowledgements
This work was supported by Cancer Research UK grant A21140 (awarded to AV). We would like to thank the Core Services and Advanced Technologies at the Cancer Research UK Beatson Institute (A17196), with particular thanks to the Metabolomics and the Cancer Research UK Glasgow Centre (A18076). We thank Catherine Winchester for helpful comments about the paper.

## Author contributions
M.P. and A.V. conceived the project. M.P. performed the in vitro experiments. G.B.B. and N.W. performed the in vivo experiments under the supervision of K.J.P. J.T.M. contributed to the in vitro experiments. D.S. and G.M.M. contributed to the untargeted and targeted metabolomics analysis.

## Competing interests
The authors declare no competing interests.
