## [Peer Review File · Communications Chemistry]

Reviewers' comments:

Reviewer #1 (Remarks to the Author):

The manuscript by Vazquez et al describes mass spectrometry analyses on the quantity and fate of formaldehyde in cells and mice. The work identifies formaldehyde concentrations using a scavenging approach coupled to LC/MS, while the authors also report identification of formaldehyde-derived adducts with amino acids. Concentrations of the identified adducts appear elevated on treatment with a formaldehyde precursor and/or when a key formaldehyde-metabolising enzyme is compromised.

Overall, this is a nice study that gives new insight into the complex biological chemistry of formaldehyde and related electrophilic metabolites. The LC/MS method is novel, although there is significant overlap with reported mass spectrometry methods, which should be better cited. However, I have some concerns with the manuscript that need addressing before publication. Firstly, I am disappointed to see that the authors have not acknowledged the recent Communications Chemistry publication from Kamps et al, which details comprehensive analyses of formaldehyde's reactions with amino acids. In this paper, formation of spinacine and timonacic (or thioproline) are detailed, including kinetic and stability studies, while N-formylcysteine and N-methylglutamate were not observed. The authors should be referring to this work in their results and discussion, in particular with respect to the stability of timonacic (for which the authors report surprisingly fast degradation rates). More detailed stability/reversibility studies should be provided for timonacic (e.g. on dilution, addition of formaldehyde scavengers, pH dependence) and the results should be compared to reported work. Secondly, the identification of N-formylcysteine is interesting as it suggests a novel formaldehyde metabolism pathway in cells. However, it is unclear whether it is derived from timonacic – the authors should highlight this uncertainty in the discussion. Finally, as the authors note, there is much debate as to the concentration of cellular formaldehyde, and therefore, it is important to discuss the other methods currently available and their limitations. Please include more comparison of this method over reported GC/MS and chromatographic methods, particularly with respect to sensitivity.

Reviewer #2 (Remarks to the Author):

Authors suggested a new way of formaldehyde utilization in mice that depends on amino acids, and support it with data from in vitro and in vivo experiments. Although formaldehyde was previously reported to form adducts with amino acids in solution, its in vivo reactivity remained unexplored, and this work fill that knowledge gap. Authors implemented a new protocol to measure formaldehyde turn-over in cell culture using methylamine as its source and demonstrated its validity. Authors than measured metabolites in cell cultures after methylamine exposure, and confirmed accumulation of timonacic, N-methyl-glutamate and spinacine in both, wild type and ADH5 -/- cells. This observation was further confirmed by 13C incorporation in cells and in mouse models in four tissues. Overall, the experimental design and results reported in this paper are of high quality and interest to the researchers in the field of one-carbon metabolism. This work very likely to be followed by numerous studies which will further explore amino-acid-dependent formaldehyde metabolism in animal models and diseases.

Despite all obvious advantages of the work, the manuscript is hard to read and follow author's way of thoughts.

In the introduction it is essential to add information about ALDH2 and catalase as another enzymes known to oxidize formaldehyde, and discuss why they're not relevant to this study (e.g., high Km of ALDH2 toward formaldehyde).

Cell experiments as presented challenge the role of glutathione in metabolism of formaldehyde, which is widely reported to be crucial for formaldehyde utilization through ADH5. That is very

unexpected result, thus needed to be discussed more in the paper. For example, what is happening with S-hydroxymethyl-GSH and S-formyl-GSH in mice tissues after MeOH treatment in both models? Is it comparable in concentrations to formaldehyde adducts with amino acids? How different are endogenous levels of GSH, cysteine and histidine in healthy cells and tissues?

There is also a consistent difference between behavior of metabolites in cells and tissues: formaldehyde adducts accumulate in WT cells, but not in WT animals after treatments. Please discuss possible reasons for that.

Such a dramatic accumulation of N-formyl-Cysteine in ADH5 deficient mice may indicate that ADH5 may directly metabolize it. Please discuss it as well.

It would be very helpful if authors update figures according to the suggestions below.

In Figure 1 please:

- 1) illustrate cell experiments in Figure 1E-H, particularly by adding reaction that methylamine converted into formaldehyde by serum SSAO.
- 2) indicate isotopes on vertical axis of 1E, 1G, and 1H, if applicable.
- 3) add to 1E-H plots substrates that were used
- 4) indicate on 1F and 1G plots that cell culture media was used.
- 5) please add the name of inhibitor or its target to 1G (e.g. SSAO inhibitor)

In Figure 2 please:

- 1) label formaldehyde, formate, N-formyl-Cysteine, S-hydroxymethyl-GSH and S-formyl-GSH in 2A if any present.
- 2) scale 2B-2E so the metabolites level could be easily compared
- 3) indicate method of metabolite measurement
- 4) add plots for peakareas of N-formyl-Cysteine, S-hydroxymethyl-GSH and S-formyl-GSH, if data available, as it is done in 2B-E.

In Figure 3 please:

- 1) illustrate experiments in solution by scheme
- 2) move 3H to a separate figure and add scheme to illustrate it
- 3) indicate isotope in 2F
- 4) add delta symbol to ADH5 in x-axis of 3H
- 5) add measurements for S-hydroxymethyl-GSH and S-formyl-GSH in 3H
- 6) add method of metabolite measurement to the legend
- 7) add p-values for every two combinations, where significant in 3H
- 8) scale all plots in 3H so the metabolites level could be easily compared.

In Figure 4 please:

- 1) add scheme to illustrate mice experiment (+ indicate age, sex)
- 2) label vertical axis as peak area/mg or /40 ml
- 3) add method of metabolite measurement to the legend
- 4) add measurements for S-hydroxymethyl-GSH and S-formyl-GSH
- 5) add p-values for every two combinations, where significant
- 6) change CTRL to WT
- 7) scale all plots so the metabolites level could be easily compared.

Also, please remove Fig 1I reference in Methods, and remove 500 rcf or 500 g in Cell Culture and design ... of Methods.

Reviewer #3 (Remarks to the Author):

This study demonstrated that excessive formaldehyde could have reaction with amino acid. However, this manuscript did not clearly indicate the pathological roles of these chemical reaction

in cancer or Alzheimer's disease. In the discussion section, the author should add some sentences to explain what was the significance of their findings.

Reviewers' comments:

Reviewer #1 (Remarks to the Author):

Firstly, I am disappointed to see that the authors have not acknowledged the recent Communications Chemistry publication from Kamps et al, which details comprehensive analyses of formaldehyde's reactions with amino acids. In this paper, formation of spinacine and timonac (or thioproline) are detailed, including kinetic and stability studies, while N-formylcysteine and N-methylglutamate were not observed. The authors should be referring to this work in their results and discussion, in particular with respect to the stability of timonac (for which the authors report surprisingly fast degradation rates).

Response: Our apologies for missing that important reference. We now discussed our results in the light of the work of Kamps et al at several points in the manuscript.

More detailed stability/reversibility studies should be provided for timonac (e.g. on dilution, addition of formaldehyde scavengers, pH dependence) and the results should be compared to reported work.

Response: Based on the experiments reported by Kamps et al, timonac is more stable at pH 7.0. In our experiments, in PBS solution at pH 7.4, it takes 24 hours to fully convert ^{12}C -timonac to ^{13}C -timonac. In contrast, in a water solution at pH 7.0, the conversion of ^{12}C -timonac to ^{13}C -timonac is not complete at 48 hours (Kamps et al). Nevertheless, in both conditions there is reversible release of formaldehyde from timonac, which is the main point we make.

Secondly, the identification of N-formylcysteine is interesting as it suggests a novel formaldehyde metabolism pathway in cells. However, it is unclear whether it is derived from timonac – the authors should highlight this uncertainty in the discussion.

Response: We agree and we have added this comment in the Discussion.

Finally, as the authors note, there is much debate as to the concentration of cellular formaldehyde, and therefore, it is important to discuss the other methods currently available and their limitations. Please include more comparison of this method over reported GC/MS and chromatographic methods, particularly with respect to sensitivity.

Response: We agree with the reviewer. We have expanded the discussion about formaldehyde concentrations in blood and tissues.

Reviewer #2 (Remarks to the Author):

In the introduction it is essential to add information about ALDH2 and catalase as another

enzymes known to oxidize formaldehyde, and discuss why they're not relevant to this study (e.g., high K_m of ALDH2 toward formaldehyde).

Response: We agree with the reviewer that the relevance of ALDH2 should be included in the introduction. With regard to catalase, we are not sure it is relevant to the turnover of formaldehyde. With regard to the activity of catalase, in the section where we feed ^{13}C -Methanol to HAP1 cells we now mention the lack of oxidation of methanol and therefore the lack of methanol oxidation by alcohol dehydrogenase or catalase in HAP1 cells.

Cell experiments as presented challenge the role of glutathione in metabolism of formaldehyde, which is widely reported to be crucial for formaldehyde utilization through ADH5. That is very unexpected result, thus needed to be discussed more in the paper. For example, what is happening with S-hydroxymethyl-GSH and S-formyl-GSH in mice tissues after MeOH treatment in both models? Is it comparable in concentrations to formaldehyde adducts with amino acids? How different are endogenous levels of GSH, cysteine and histidine in healthy cells and tissues?

Response: These experiments do not challenge the role of glutathione in metabolism of formaldehyde. As we have shown previously, ADH5 competent cells convert formaldehyde to formate but ADH5 knockout cells cannot (Burgos-Barragan et al 2017), cited in the text. What our data suggest is that there is a parallel pathway of formaldehyde metabolism associated with its reaction with amino acids. We have added this clarification in the discussion.

There is also a consistent difference between behavior of metabolites in cells and tissues: formaldehyde adducts accumulate in WT cells, but not in WT animals after treatments. Please discuss possible reasons for that.

Such a dramatic accumulation of N-formyl-Cysteine in ADH5 deficient mice may indicate that ADH5 may directly metabolize it. Please discuss it as well.

Response: We agree with the reviewer. We have added a discussion about the differences between the in vitro and in vivo data at the end of the Results section. The accumulation of N-formylcysteine could be also the consequence of increased production of N-formylcysteine rather than decreased turnover. We have also discussed this point.

In Figure 1 please:

- 1) illustrate cell experiments in Figure 1E-H, particularly by adding reaction that methylamine converted into formaldehyde by serum SSAO.
- 2) indicate isotopes on vertical axis of 1E, 1G, and 1H, if applicable.
- 3) add to 1E-H plots substrates that were used
- 4) indicate on 1F and 1G plots that cell culture media was used.
- 5) please add the name of inhibitor or its target to 1G (e.g. SSAO inhibitor)

Response: 1) These experiments have different designs. Including illustrations will overload the figure. We have included panel labels (on top of each panel) to guide the reader. 2) Not applicable. 3) Done. 4) Done. 5) Done.

In Figure 2 please:

- 1) label formaldehyde, formate, N-formyl-Cysteine, S-hydroxymethyl-GSH and S-formyl-GSH in 2A if any present.
- 2) scale 2B-2E so the metabolites level could be easily compared
- 3) indicate method of metabolite measurement
- 4) add plots for peakareas of N-formyl-Cysteine, S-hydroxymethyl-GSH and S-formyl-GSH, if data available, as it is done in 2B-E.

Response: 1) These compounds were not identified by the untargeted analysis. 2) Peak areas of different compounds cannot be compared because different compounds have different ionization efficiencies. 3) We have added an explicit mention to the method used in the caption. 4) These compounds were not identified by the untargeted analysis.

In Figure 3 please:

- 1) illustrate experiments in solution by scheme
- 2) move 3H to a separate figure and add scheme to illustrate it
- 3) indicate isotope in 2F
- 4) add delta symbol to ADH5 in x-axis of 3H
- 5) add measurements for S-hydroxymethyl-GSH and S-formyl-GSH in 3H
- 6) add method of metabolite measurement to the legend
- 7) add p-values for every two combinations, where significant in 3H
- 8) scale all plots in 3H so the metabolites level could be easily compared.

Response: 1) Described in the text. 2) We prefer to keep this data in the same figure. 3) Not applicable, as this is intracellular GSH not reacting with the ¹³C-substrate. 4) Here ADH5 stands for HAP1-DeltaADH5, which is too large to be displayed in the figure. We have specified in the caption that ADH5 stands for HAP1-DeltaADH5. 5) Not detected. 6) Reported in the Methods. 7) We add p-values only when relevant for the discussion of the data in the text. 8) Peak areas of different compounds cannot be compared because different compounds have different ionization efficiencies.

In Figure 4 please:

- 1) add scheme to illustrate mice experiment (+ indicate age, sex)
- 2) label vertical axis as peak area/mg or /40 ml
- 3) add method of metabolite measurement to the legend
- 4) add measurements for S-hydroxymethyl-GSH and S-formyl-GSH
- 5) add p-values for every two combinations, where significant

6) change CTRL to WT

7) scale all plots so the metabolites level could be easily compared.

Response: 1) We indicated that mice had mixed age and gender in the caption 2) The Y-axis has been labelled accordingly 3) Done. 4) Not detected reliably. 5) We add p-values only when relevant for the discussion of the data in the text. 6) CTR changed to WT. 7) Peak areas of different compounds cannot be compared because different compounds have different ionization efficiencies.

Also, please remove Fig 1l reference in Methods, and remove 500 rcf or 500 g in Cell Culture and design ... of Methods.

Response: Done.

Reviewer #3 (Remarks to the Author):

This study demonstrated that excessive formaldehyde could have reaction with amino acid. However, this manuscript did not clearly indicate the pathological roles of these chemical reaction in cancer or Alzheimer's disease. In the discussion section, the author should add some sentences to explain what was the significance of their findings.

Response: We have added a discussion on the relevance of these observations to human disease.

REVIEWERS' COMMENTS:

Reviewer #2 (Remarks to the Author):

All my points raised in the previous round of review have been satisfactorily addressed. I recommend the paper for publication.